# Global Sparse Momentum SGD for Pruning Very Deep Neural Networks

**Xiaohan Ding** [1]    **Guiguang Ding** [1]    **Xiangxin Zhou** [2]
**Yuchen Guo** [1,3]    **Jungong Han** [4]    **Ji Liu** [5]

[1] Beijing National Research Center for Information Science and Technology (BNRist);
School of Software, Tsinghua University, Beijing, China
[2] Department of Electronic Engineering, Tsinghua University, Beijing, China
[3] Department of Automation, Tsinghua University;
Institute for Brain and Cognitive Sciences, Tsinghua University, Beijing, China
[4] WMG Data Science, University of Warwick, Coventry, United Kingdom
[5] Kwai Seattle AI Lab, Kwai FeDA Lab, Kwai AI platform
dxh17@mails.tsinghua.edu.cn    dinggg@tsinghua.edu.cn
xx-zhou16@mails.tsinghua.edu.cn    yuchen.w.guo@gmail.com
jungonghan77@gmail.com    ji.liu.uwisc@gmail.com

## Abstract

Deep Neural Network (DNN) is powerful but computationally expensive and memory intensive, thus impeding its practical usage on resource-constrained front-end devices. DNN pruning is an approach for deep model compression, which aims at eliminating some parameters with tolerable performance degradation. In this paper, we propose a novel momentum-SGD-based optimization method to reduce the network complexity by on-the-fly pruning. Concretely, given a global compression ratio, we categorize all the parameters into two parts at each training iteration which are updated using different rules. In this way, we gradually zero out the redundant parameters, as we update them using only the ordinary weight decay but no gradients derived from the objective function. As a departure from prior methods that require heavy human works to tune the layer-wise sparsity ratios, prune by solving complicated non-differentiable problems or finetune the model after pruning, our method is characterized by 1) global compression that automatically finds the appropriate per-layer sparsity ratios; 2) end-to-end training; 3) no need for a time-consuming re-training process after pruning; and 4) superior capability to find better winning tickets which have won the initialization lottery.

## 1   Introduction

The recent years have witnessed great success of Deep Neural Network (DNN) in many real-world applications. However, today's very deep models have been accompanied by millions of parameters, thus making them difficult to be deployed on computationally limited devices. In this context, DNN pruning approaches have attracted much attention, where we eliminate some connections (i.e., individual parameters) [21, 22, 31], or channels [32], thus the required storage space and computations can be reduced. This paper is focused on connection pruning, but the proposed method can be easily generalized to structured pruning (e.g., neuron-, kernel- or filter-level). In order to reach a good trade-off between accuracy and model size, many pruning methods have been proposed, which can be categorized into two typical paradigms. **1)** Some researchers [13, 18, 21, 22, 26, 31, 32, 39, 41] propose to prune the model by some means to reach a certain level of compression ratio, then finetune it using ordinary SGD to restore the accuracy. **2)** The other methods seek to produce sparsity in the model through a customized learning procedure [1, 12, 33, 34, 51, 54, 56].

Though the existing methods have achieved great success in pruning, there are some typical drawbacks. Specifically, when we seek to prune a model in advance and finetune it, we confront two problems:

- The layer-wise sparsity ratios are inherently tricky to set as hyper-parameters. Many previous works [21, 24, 26, 32] have shown that some layers in a DNN are sensitive to pruning, but some can be pruned significantly without degrading the model in accuracy. As a consequence, it requires prior knowledge to tune the layer-wise hyper-parameters in order to maximize the global compression ratio without unacceptable accuracy drop.
- The pruned models are difficult to train, and we cannot predict the final accuracy after finetuning. E.g., the filter-level-pruned models can be easily trapped into a bad local minima, and sometimes cannot even reach a similar level of accuracy with a counterpart trained from scratch [10, 38]. And in the context of connection pruning, the sparser the network, the slower the learning and the lower the eventual test accuracy [15].

On the other hand, pruning by learning is not easier due to:

- In some cases we introduce a hyper-parameter to control the trade-off, which does not directly reflect the resulting compression ratio. For instance, MorphNet [17] uses group Lasso [44] to zero out some filters for structured pruning, where a key hyper-parameter is the Lasso coefficient. However, given a specific value of the coefficient, we cannot predict the final compression ratio before the training ends. Therefore, when we target at a specific eventual compression ratio, we have to try multiple coefficient values in advance and choose the one that yields the result closest to our expectation.
- Some methods prune by solving an optimization problem which directly concerns the sparsity. As the problem is non-differentiable, it cannot be solved using SGD-based methods in an end-to-end manner. A more detailed discussion will be provided in Sect. 3.2.

In this paper, we seek to overcome the drawbacks discussed above by directly altering the gradient flow based on momentum SGD, which explicitly concerns the eventual compression ratio and can be implemented via end-to-end training. Concretely, we use first-order Taylor series to measure the importance of a parameter by estimating how much the objective function value will be changed by removing it [41, 49]. Based on that, given a global compression ratio, we categorize all the parameters into two parts that will be updated using different rules, which is referred to as *activation selection*. For the unimportant parameters, we perform *passive update* with no gradients derived from the objective function but only the ordinary weight decay (i.e., $\ell$-2 regularization) to penalize their values. On the other hand, via *active update*, the critical parameters are updated using both the objective-function-related gradients and the weight decay to maintain the model accuracy. Such a selection is conducted at each training iteration, so that a deactivated connection gets a chance to be reactivated at the next iteration. Through continuous *momentum-accelerated* passive updates we can make most of the parameters infinitely close to zero, such that pruning them causes no damage to the model's accuracy. Owing to this, there is no need for a finetuning process. In contrast, some previously proposed regularization terms can only reduce the parameters to some extent, thus pruning still degrades the model. Our contributions are summarized as follows.

- For lossless pruning and end-to-end training, we propose to directly alter the gradient flow, which is clearly distinguished with existing methods that either add a regularization term or seek to solve some non-differentiable optimization problems.
- We propose Global Sparse Momentum SGD (GSM), a novel SGD optimization method, which splits the update rule of momentum SGD into two parts. GSM-based DNN pruning requires a sole global eventual compression ratio as hyper-parameter and can automatically discover the appropriate per-layer sparsity ratios to achieve it.
- Seen from the experiments, we have validated the capability of GSM to achieve high compression ratios on MNIST, CIFAR-10 [29] and ImageNet [9] as well as find better winning tickets [15]. The codes are available at `https://github.com/DingXiaoH/GSM-SGD`.

## 2   Related work

### 2.1   Momentum SGD

Stochastic gradient descent only takes the first order derivatives of the objective function into account and not the higher ones [28]. Momentum is a popular technique used along with SGD, which

accumulates the gradients of the past steps to determine the direction to go, instead of using only the gradient of the current step. I.e., momentum gives SGD a short-term memory [16]. Formally, let $L$ be the objective function, $w$ be a single parameter, $\alpha$ be the learning rate, $\beta$ be the momentum coefficient which controls the percentage of the gradient retained every iteration, $\eta$ be the ordinary weight decay coefficient (e.g., $1 \times 10^{-4}$ for ResNets [23]), the update rule is

$$z^{(k+1)} \leftarrow \beta z^{(k)} + \eta w^{(k)} + \frac{\partial L}{\partial w^{(k)}} \,,$$
$$w^{(k+1)} \leftarrow w^{(k)} - \alpha z^{(k+1)} \,. \tag{1}$$

There is a popular story about momentum [16, 42, 45, 48]: gradient descent is a man walking down a hill. He follows the steepest path downwards; his progress is slow, but steady. Momentum is a heavy ball rolling down the same hill. The added inertia acts both as a smoother and an accelerator, dampening oscillations and causing us to barrel through narrow valleys, small humps and local minima. In this paper, we use momentum as an accelerator to boost the passive updates.

## 2.2 DNN pruning and other techniques for compression and acceleration

DNN pruning seeks to remove some parameters without significant accuracy drop, which can be categorized into unstructured and structured techniques based on the pruning granularity. Unstructured pruning (a.k.a. connection pruning) [7, 21, 22, 31] targets at significantly reducing the number of non-zero parameters, resulting in a sparse model, which can be stored using much less space, but cannot effectively reduce the computational burdens on off-the-shelf hardware and software platforms. On the other hand, structured pruning removes structures (e.g., neurons, kernels or whole filters) from DNN to obtain practical speedup. E.g., channel pruning [10, 11, 32, 35, 37, 38] cannot achieve an extremely high compression ratio of the model size, but can convert a wide CNN into a narrower (but still dense) one to reduce the memory and computational costs. In real-world applications, unstructured and structured pruning are often used together to achieve the desired trade-off.

This paper is focused on connection pruning (but the proposed method can be easily generalized to structured pruning), which has attracted much attention since Han et al. [21] pruned DNN connections based on the magnitude of parameters and restored the accuracy via ordinary SGD. Some inspiring works have improved the paradigm of pruning-and-finetuning by splicing connections as they become important again [18], directly targeting at the energy consumption [55], utilizing per-layer second derivatives [13], etc. The other learning-based pruning methods will be discussed in Sect. 3.2

Apart from pruning, we can also compress and accelerate DNN in other ways. Some works [2, 46, 57] decompose or approximate parameter tensors; quantization and binarization techniques [8, 19, 20, 36] approximate a model using fewer bits per parameter; knowledge distillation [3, 25, 43] transfers knowledge from a big network to a smaller one; some researchers seek to speed up convolution with the help of perforation [14], FFT [40, 50] or DCT [53]; Wang et al. [52] compact feature maps by extracting information via a Circulant matrix.

## 3 GSM: Global Sparse Momentum SGD

### 3.1 Formulation

We first clarify the notations in this paper. For a fully-connected layer with $p$-dimensional input and $q$-dimensional output, we use $\boldsymbol{W} \in \mathbb{R}^{p \times q}$ to denote the kernel matrix. For a convolutional layer with kernel tensor $\boldsymbol{K} \in \mathbb{R}^{h \times w \times r \times s}$, where $h$ and $w$ are the height and width of convolution kernel, $r$ and $s$ are the numbers of input and output channels, respectively, we unfold the tensor $\boldsymbol{K}$ into $\boldsymbol{W} \in \mathbb{R}^{hwr \times s}$. Let $N$ be the number of all such layers, we use $\boldsymbol{\Theta} = [\boldsymbol{W}_i]$ ($\forall 1 \le i \le N$) to denote the collection of all such kernel matrices, and the global compression ratio $C$ is given by

$$C = \frac{|\boldsymbol{\Theta}|}{||\boldsymbol{\Theta}||_0} \,, \tag{2}$$

where $|\boldsymbol{\Theta}|$ is the size of $\boldsymbol{\Theta}$ and $||\boldsymbol{\Theta}||_0$ is the $\ell$-0 norm, i.e., the number of non-zero entries. Let $L$, $X$, $Y$ be the accuracy-related loss function (e.g., cross entropy for classification tasks), test examples and labels, respectively, we seek to obtain a good trade-off between accuracy and model size by achieving a high compression ratio $C$ without unacceptable increase in the loss $L(X, Y, \boldsymbol{\Theta})$.

## 3.2 Rethinking learning-based pruning

The optimization target or direction of ordinary DNN training is to minimize the objective function only, but when we seek to produce a sparse model via a customized learning procedure, the key is to *deviate the original training direction* by taking into account the sparsity of the parameters. Through training, the sparsity emerges progressively, and we eventually reach the expected trade-off between accuracy and model size, which is usually controlled by one or a series of hyper-parameters.

### 3.2.1 Explicit trade-off as constrained optimization

The trade-off can be explicitly modeled as a constrained optimization problem [56], e.g.,

$$\underset{\Theta}{\text{minimize}} \quad L(X, Y, \Theta) + \sum_{i=1}^{N} g_i(\boldsymbol{W}_i) , \tag{3}$$

where $g_i$ is an indicator function,

$$g_i(\boldsymbol{W}) = \begin{cases} 0 & \text{if } ||\boldsymbol{W}||_0 \leq l_i , \\ +\infty & \text{otherwise} , \end{cases} \tag{4}$$

and $l_i$ is the required number of non-zero parameters at layer $i$. Since the second term of the objective function is non-differentiable, the problem cannot be settled analytically or by stochastic gradient descent, but can be tackled by alternately applying SGD and solving the non-differentiable problem, e.g., using ADMM [6]. In this way, the training direction is deviated, and the trade-off is obtained.

### 3.2.2 Implicit trade-off using regularizations

It is a common practice to apply some extra differentiable regularizations during training to reduce the magnitude of some parameters, such that removing them causes less damage [1, 21, 54]. Let $R(\Theta)$ be the magnitude-related regularization term, $\lambda$ be a trade-off hyper-parameter, the problem is

$$\underset{\Theta}{\text{minimize}} \quad L(X, Y, \Theta) + \lambda R(\Theta) . \tag{5}$$

However, the weaknesses are two-fold. **1)** Some common regularizations, e.g., $\ell$-1, $\ell$-2 and Lasso [44], cannot literally zero out the entries in $\Theta$, but can only reduce the magnitude to some extent, such that removing them still degrades the performance. We refer to this phenomenon as the *magnitude plateau*. The cause behind is simple: for a specific trainable parameter $w$, when its magnitude $|w|$ is large at the beginning, the gradient derived from $R$, i.e., $\lambda \frac{\partial R}{\partial w}$, overwhelms $\frac{\partial L}{\partial w}$, thus $|w|$ is gradually reduced. However, as $|w|$ shrinks, $\frac{\partial R}{\partial w}$ diminishes, too, such that the reducing tendency of $|w|$ plateaus when $\frac{\partial R}{\partial w}$ approaches $\frac{\partial L}{\partial w}$, and $w$ maintains a relatively small magnitude. **2)** The hyper-parameter $\lambda$ does not directly reflect the resulting compression ratio, thus we may need to make several attempts to gain some empirical knowledge before we obtain the model with our expected compression ratio.

## 3.3 Global sparse gradient flow via momentum SGD

To overcome the drawbacks of the two paradigms discussed above, we intend to explicitly control the eventual compression ratio via end-to-end training by directly altering the gradient flow of momentum SGD to deviate the training direction in order to achieve a high compression ratio as well as maintain the accuracy. Intuitively, we seek to use the gradients to guide the few active parameters in order to minimize the objective function, and penalize most of the parameters to push them *infinitely close to zero*. Therefore, the first thing is to find a proper metric to distinguish the active part. Given a global compression ratio $C$, we use $Q = \frac{|\Theta|}{C}$ to denote the number of non-zero entries in $\Theta$. At each training iteration, we feed a mini-batch of data into the model, compute the gradients using the ordinary chain rule, calculate the metric values for every parameter, perform active update on $Q$ parameters with the largest metric values and passive update on the others. In order to make GSM feasible on very deep models, the metrics should be calculated using only the original intermediate computational results, i.e., the parameters and gradients, but no second-order derivatives. Inspired by two preceding methods which utilized first-order Taylor series for greedy channel pruning [41, 49],

we define the metric in a similar manner. Formally, at each training iteration with a mini-batch of examples $x$ and labels $y$, let $T(x, y, w)$ be the metric value of a specific parameter $w$, we have

$$T(x, y, w) = \left| \frac{\partial L(x, y, \boldsymbol{\Theta})}{\partial w} w \right| . \tag{6}$$

The theory is that for the current mini-batch, we expect to reduce those parameters which can be removed with less impact on $L(x, y, \boldsymbol{\Theta})$. Using the Taylor series, if we set a specific parameter $w$ to 0, the loss value becomes

$$L(x, y, \boldsymbol{\Theta}_{w \leftarrow 0}) = L(x, y, \boldsymbol{\Theta}) - \frac{\partial L(x, y, \boldsymbol{\Theta})}{\partial w}(0 - w) + o(w^2) . \tag{7}$$

Ignoring the higher-order term, we have

$$|L(x, y, \boldsymbol{\Theta}_{w \leftarrow 0}) - L(x, y, \boldsymbol{\Theta})| = \left| \frac{\partial L(x, y, \boldsymbol{\Theta})}{\partial w} w \right| = T(x, y, w) , \tag{8}$$

which is an approximation of the change in the loss value if $w$ is zeroed out.

We rewrite the update rule of momentum SGD (Formula 1). At the $k$-th training iteration with a mini-batch of examples $x$ and labels $y$ on a specific layer with kernel $\boldsymbol{W}$, the update rule is

$$\begin{aligned}
\boldsymbol{Z}^{(k+1)} &\leftarrow \beta \boldsymbol{Z}^{(k)} + \eta \boldsymbol{W}^{(k)} + \boldsymbol{B}^{(k)} \circ \frac{\partial L(x, y, \boldsymbol{\Theta})}{\partial \boldsymbol{W}^{(k)}} , \\
\boldsymbol{W}^{(k+1)} &\leftarrow \boldsymbol{W}^{(k)} - \alpha \boldsymbol{Z}^{(k+1)} ,
\end{aligned} \tag{9}$$

where $\circ$ is the element-wise multiplication (a.k.a. Hadamard-product), and $\boldsymbol{B}^{(k)}$ is the mask matrix,

$$B_{m,n}^{(k)} = \begin{cases} 1 & \text{if } T(x, y, W_{m,n}^{(k)}) \geq \text{the } Q\text{-th greatest value in } T(x, y, \boldsymbol{\Theta}^{(k)}) , \\ 0 & \text{otherwise} . \end{cases} \tag{10}$$

We refer to the computation of $\boldsymbol{B}$ for each kernel as *activation selection*. Obviously, there are exactly $Q$ ones in all the mask matrices, and GSM degrades to ordinary momentum SGD when $Q = |\boldsymbol{\Theta}|$.

Of note is that GSM is model-agnostic because it makes no assumptions on the model structure or the form of loss function. I.e., the calculation of gradients via back propagation is model-related, of course, but it is model-agnostic to use them for GSM pruning.

### 3.4 GSM enables implicit reactivation and fast continuous reduction

As GSM conducts activation selection at each training iteration, it allows the penalized connections to be reactivated, if they are found to be critical to the model again. Compared to two previous works which explicitly insert a splicing [18] or restoring [55] stage into the entire pipeline to rewire the mistakenly pruned connections, GSM features simpler implementation and end-to-end training.

However, as will be shown in Sect. 4.4, re-activation only happens on a minority of the parameters, but most of them undergo a series of passive updates, thus keep moving towards zero. As we would like to know how many training iterations are needed to make the parameters small enough to realize lossless pruning, we need to predict the eventual value of a parameter $w$ after $k$ passive updates, given $\alpha$, $\eta$ and $\beta$. We can use Formula 1 to predict $w^{(k)}$, which is practical but cumbersome. In our common use cases where $z^{(0)} = 0$ (from the very beginning of training), $k$ is large (at least tens of thousands), and $\alpha\eta$ is small (e.g., $\alpha = 5 \times 10^{-3}$, $\eta = 5 \times 10^{-4}$), we have observed an empirical formula which is precise enough (Fig. 1) to approximate the resulting value,

$$\frac{w^{(k)}}{w^{(0)}} \approx (1 - \frac{\alpha\eta}{1 - \beta})^k . \tag{11}$$

In practice, we fix $\eta$ (e.g., $1 \times 10^{-4}$ for ResNets [23] and DenseNets [27]) and adjust $\alpha$ just as we do for ordinary DNN training, and use $\beta = 0.98$ or $\beta = 0.99$ for $50\times$ or $100\times$ faster zeroing-out. When the training is completed, we prune the model globally by only preserving $Q$ parameters in $\boldsymbol{\Theta}$ with the largest magnitude. We decide the number of training iterations $k$ using Eq. 11 based

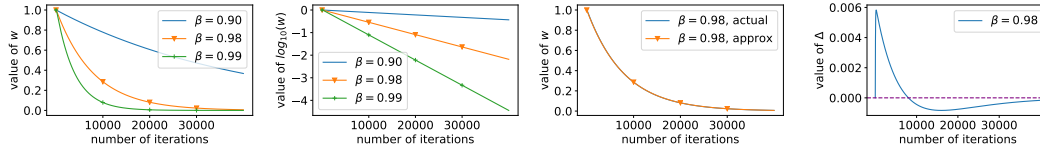

Figure 1: Value of a parameter $w$ after continuous passive updates with $\alpha = 5 \times 10^{-3}$, $\eta = 5 \times 10^{-4}$, assuming $w^{(0)} = 1$. First and second figures: the actual value of $w$ obtained using Formula 1 with different momentum coefficient $\beta$. Note the logarithmic scale of the second figure. Clearly, a larger $\beta$ can accelerate the reduction of parameter value. Third and fourth figures: the value approximated by Eq. 11 and the difference $\Delta = w_{actual} - w_{approx}$ with $\beta = 0.98$ as the representative.

on an empirical observation that with $(1 - \frac{\alpha\eta}{1-\beta})^k < 1 \times 10^{-4}$, such a pruning operation causes no accuracy drop on very deep models like ResNet-56 and DenseNet-40.

Momentum is critical for GSM-based pruning to be completed with acceptable time cost. As most of the parameters continuously grow in the same direction determined by the weight decay (i.e., towards zero), such a tendency accumulates in the momentum, thus the zeroing-out process is significantly accelerated. On the other hand, if a parameter does not always vary in the same direction, raising $\beta$ less affect its training dynamics. In contrast, if we increase the learning rate $\alpha$ for faster zeroing-out, the critical parameters which are hovering around the global minima will significantly deviate from their current values reached with a much lower learning rate before.

# 4 Experiments

## 4.1 Pruning results and comparisons

We evaluate GSM by pruning several common benchmark models on MNIST, CIFAR-10 [29] and ImageNet [9], and comparing with the reported results from several recent competitors. For each trial, we start from a well-trained base model and apply GSM training on all the layers *simultaneously*.

**MNIST.** We first experiment on MNIST with LeNet-300-100 and LeNet-5 [30]. LeNet-300-100 is a three-layer fully-connected network with 267K parameters, which achieves 98.19% Top1 accuracy. LeNet-5 is a convolutional network which comprises two convolutional layers and two fully-connected layers, contains 431K parameters and delivers 99.21% Top1 accuracy. To achieve $60\times$ and $125\times$ compression, we set $Q = \frac{267K}{60} = 4.4K$ for LeNet-300-100 and $Q = \frac{431K}{125} = 3.4K$ for LeNet-5, respectively. We use momentum coefficient $\beta = 0.99$ and a batch size of 256. The learning rate schedule is $\alpha = 3 \times 10^{-2}, 3 \times 10^{-3}, 3 \times 10^{-4}$ for 160, 40 and 40 epochs, respectively. After GSM training, we conduct lossless pruning and test on the validation dataset. As shown in Table. 1, GSM can produce highly sparse models which still maintain the accuracy. By further raising the compression ratio on LeNet-5 to $300\times$, we only observe a minor accuracy drop (0.15%), which suggests that GSM can yield reasonable performance with extremely high compression ratios.

**CIFAR-10.** We present the results of another set of experiments on CIFAR-10 in Table. 2 using ResNet-56 [23] and DenseNet-40 [27]. We use $\beta = 0.98$, a batch size of 64 and learning rate $\alpha = 5 \times 10^{-3}, 5 \times 10^{-4}, 5 \times 10^{-5}$ for 400, 100 and 100 epochs, respectively. We adopt the standard data augmentation including padding to $40 \times 40$, random cropping and left-right flipping. Though ResNet-56 and DenseNet-40 are significantly deeper and more complicated, GSM can also reduce the parameters by $10\times$ and still maintain the accuracy.

**ImageNet.** We prune ResNet-50 to verify GSM on large-scale image recognition applications. We use a batch size of 64 and train the model with $\alpha = 1 \times 10^{-3}, 1 \times 10^{-4}, 1 \times 10^{-5}$ for 40, 10 and 10 epochs, respectively. We compare the results with L-OBS [13], which is the only comparable previous method that reported experimental results on ResNet-50, to the best of our knowledge. Obviously, GSM outperforms L-OBS by a clear margin (Table. 3). We assume that the effectiveness of GSM on such a very deep network is due to its capability to discover the appropriate layer-wise sparsity ratios, given a desired global compression ratio. In contrast, L-OBS performs pruning layer by layer using the same compression ratio. This assumption is further verified in Sect. 4.2.

Table 1: Pruning results on MNIST.

| Model | Result | Base Top1 | Pruned Top1 | Origin / Remain Params | Compress Ratio | Non-zero Ratio |
|-------|--------|-----------|-------------|------------------------|----------------|----------------|
| LeNet-300 | Han et al. [21] | 98.36 | 98.41 | 267K / 22K | 12.1× | 8.23% |
| LeNet-300 | L-OBS [13] | 98.24 | 98.18 | 267K / 18.6K | 14.2× | 7% |
| LeNet-300 | Zhang et al. [56] | 98.4 | 98.4 | 267K / 11.6K | 23.0× | 4.34% |
| LeNet-300 | DNS [18] | 97.72 | 98.01 | 267K / 4.8K | 55.6× | 1.79% |
| **LeNet-300** | **GSM** | **98.19** | **98.18** | **267K / 4.4K** | **60.0×** | **1.66%** |
| LeNet-5 | Han et al. [21] | 99.20 | 99.23 | 431K / 36K | 11.9× | 8.35% |
| LeNet-5 | L-OBS [13] | 98.73 | 98.73 | 431K / 3.0K | 14.1× | 7% |
| LeNet-5 | Srinivas et al. [47] | 99.20 | 99.19 | 431K / 22K | 19.5× | 5.10% |
| LeNet-5 | Zhang et al. [56] | 99.2 | 99.2 | 431K / 6.05K | 71.2× | 1.40% |
| LeNet-5 | DNS [18] | 99.09 | 99.09 | 431K / 4.0K | 107.7× | 0.92% |
| **LeNet-5** | **GSM** | **99.21** | **99.22** | **431K / 3.4K** | **125.0×** | **0.80%** |
| **LeNet-5** | **GSM** | **99.21** | **99.06** | **431K / 1.4K** | **300.0×** | **0.33%** |

Table 2: Pruning results on CIFAR-10.

| Model | Result | Base Top1 | Pruned Top1 | Origin / Remain Params | Compress Ratio | Non-zero Ratio |
|-------|--------|-----------|-------------|------------------------|----------------|----------------|
| ResNet-56 | GSM | 94.05 | 94.10 | 852K / 127K | 6.6× | 15.0% |
| ResNet-56 | GSM | 94.05 | 93.80 | 852K / 85K | 10.0× | 10.0% |
| DenseNet-40 | GSM | 93.86 | 94.07 | 1002K / 150K | 6.6× | 15.0% |
| DenseNet-40 | GSM | 93.86 | 94.02 | 1002K / 125K | 8.0× | 12.5% |
| DenseNet-40 | GSM | 93.86 | 93.90 | 1002K / 100K | 10.0× | 10.0% |

## 4.2 GSM for automatic layer-wise sparsity ratio decision

Modern DNNs usually contain tens or even hundreds of layers. As the architectures deepen, it becomes increasingly impractical to set the layer-wise sparsity ratios manually to reach a desired global compression ratio. Therefore, the research community is soliciting techniques which can automatically discover the appropriate sparsity ratios on very deep models. In practice, we noticed that if directly pruning a single layer of the original model by a fixed ratio results in a significant accuracy reduction, GSM automatically chooses to prune it less, and vice versa.

In this subsection, we present a quantitative analysis of the *sensitivity* to pruning, which is an underlying property of a layer defined via a natural proxy: the accuracy reduction caused by pruning a certain ratio of parameters from it. We first evaluate such sensitivity via single-layer pruning attempts with different pruning ratios (Fig. 2). E.g., for the curve labeled as "prune 90%" of LeNet-5, we first experiment on the first layer by setting 90% of the parameters with smaller magnitude to zero then testing on the validation set. Then we restore the first layer, prune the second layer and test. The same procedure is applied to the third and fourth layers. After that, we use different pruning ratios of 99%, 99.5%, 99.7%, and obtain three curves in the same way. From such experiments we learn that the first layer is far more sensitive than the third, as pruning 99% of the parameters from the first layer reduces the Top1 accuracy by around 85% (i.e., to hardly above 10%), but doing so on the third layer only slightly degrades the accuracy by 3%.

Then we show the resulting layer-wise non-zero ratio of the GSM-pruned models (125× pruned LeNet-5 and 6.6× pruned DenseNet-40, as reported in Table. 1, 2) as another proxy for sensitivity, of which the curves are labeled as "GSM discovered" in Fig. 2. As the two curves vary in the same

Table 3: Pruning results on ImageNet.

| Model | Result | Base Top1 / Top5 | Pruned Top1 / Top5 | Origin / Remain Params | Compress Ratio | Non-zero Ratio |
|-------|--------|------------------|--------------------|------------------------|----------------|----------------|
| ResNet-50 | L-OBS[13] | - / ≈ 92 | - / ≈ 92 | 25.5M / 16.5M | 1.5× | 65% |
| ResNet-50 | L-OBS[13] | - / ≈ 92 | - / ≈ 85 | 25.5M / 11.4M | 2.2× | 45% |
| **ResNet-50** | **GSM** | **75.72 / 92.75** | **75.33 / 92.47** | **25.5M / 6.3M** | **4.0×** | **25%** |
| **ResNet-50** | **GSM** | **75.72 / 92.75** | **74.30 / 91.98** | **25.5M / 5.1M** | **5.0×** | **20%** |

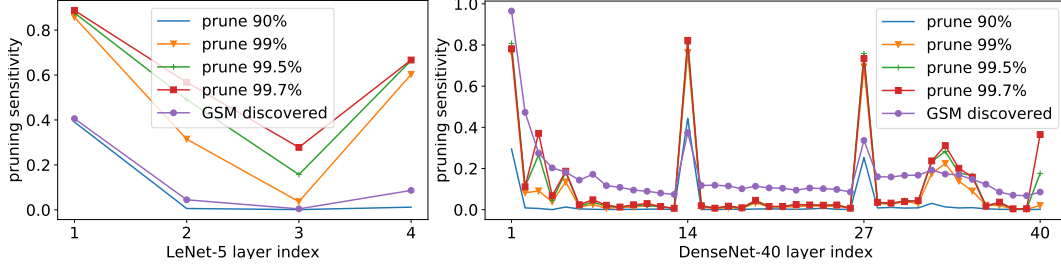

Figure 2: The layer sensitivity scores estimated by both layer-wise pruning attempts and GSM on LeNet-5 (left) and DenseNet-40 (right). Though the numeric values of the two sensitivity proxies on the same layer are not comparable, they vary in the same tendency across layers.

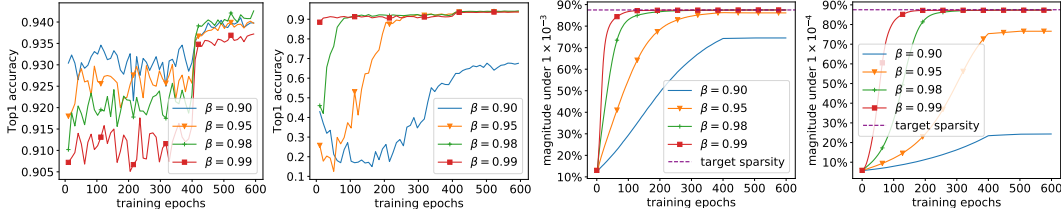

(a) Original accuracy.     (b) Pruned accuracy.     (c) Ratio under $1 \times 10^{-3}$.   (d) Ratio under $1 \times 10^{-4}$.

Figure 3: The accuracy curves obtained by evaluating both the original model and the globally $8\times$ pruned, and the ratio of parameters of which the magnitude is under $1 \times 10^{-3}$ or $1 \times 10^{-4}$, respectively, using different values of momentum coefficient $\beta$. Best viewed in color.

tendency across layers as others, we find out that the sensitivities measured in the two proxies are closely related, which suggests that GSM automatically decides to prune the sensitive layers less (e.g., the 14th, 27th and 40th layer in DenseNet-40, which perform the inter-stage transitions [27]) and the insensitive layers more in order to reach the desired global compression ratio, eliminating the need for heavy human works to tune the sparsity ratios as hyper-parameters.

### 4.3 Momentum for accelerating parameter zeroing-out

We investigate the role momentum plays in GSM by only varying the momentum coefficient $\beta$ and keeping all the other training configurations the same as the $8\times$ pruned DenseNet-40 in Sect. 4.1. During training, we evaluate the model both before and after pruning every 8000 iterations (i.e., 10.24 epochs). We also present in Fig. 3 the global ratio of parameters with magnitude under $1 \times 10^{-3}$ and $1 \times 10^{-4}$, respectively. As can be observed, a large momentum coefficient can drastically increase the ratio of small-magnitude parameters. E.g., with a target compression ratio of $8\times$ and $\beta = 0.98$, GSM can make 87.5% of the parameters close to zero (under $1 \times 10^{-4}$) in around 150 epochs, thus pruning the model causes no damage. And with $\beta = 0.90$, 400 epochs are not enough to effectively zero the parameters out, thus pruning degrades the accuracy to around 65%. On the other hand, as a larger $\beta$ value brings more rapid structural change in the model, the original accuracy decreases at the beginning but increases when such change becomes stable and the training converges.

### 4.4 GSM for implicit connection reactivation

GSM implicitly implements connection rewiring by performing activation selection at each iteration to restore the parameters which have been wrongly penalized (i.e., gone through at least one passive update). We investigate the significance of doing so by pruning DenseNet-40 by $8\times$ again using $\beta = 0.98$ and the same training configurations as before but without re-selection (Fig. 4). Concretely, we use the mask matrices computed at the first iteration to guide the updates until the end of training. It is observed that if re-selection is canceled, the training loss becomes higher, and the accuracy is degraded. This is because the first selection decides to eliminate some connections which are not critical for the first iteration but may be important for the subsequent input examples. Without re-selection, GSM insists on zeroing out such parameters, leading to lower accuracy. And by depicting the reactivation ratio (i.e., the ratio of the number of parameters which switch from passive to active to the total number of parameters) at the re-selection of each training iteration, we learn that reactivation

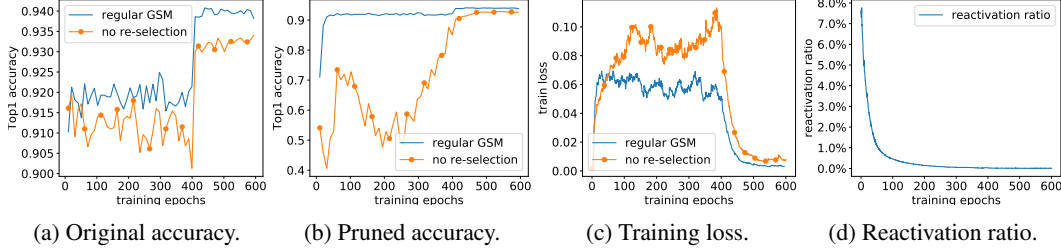

(a) Original accuracy.    (b) Pruned accuracy.    (c) Training loss.    (d) Reactivation ratio.

Figure 4: The training process of GSM both with and without re-selection.

Table 4: Eventual Top1 accuracy of the winning tickets training (step 5).

| Model | Compression ratio | Magnitude tickets | GSM tickets |
|---|---|---|---|
| LeNet-300 | 60× | 97.39 | 98.22 |
| LeNet-5 | 125× | 97.60 | 99.04 |
| LeNet-5 | 300× | 11.35 | 98.88 |

happens on a minority of the connections, and the ratio decreases gradually, such that the training converges and the desired sparsity ratio is obtained.

### 4.5 GSM for more powerful winning lottery tickets

Frankle and Carbin [15] reported that the parameters which are found to be important after training are actually important at the very beginning (after random initialization but before training), which are referred to as the winning tickets, because they have won the initialization lottery. It is discovered that if we **1)** randomly initialize a network parameterized by $\Theta_0$, **2)** train and obtain $\Theta$, **3)** prune some parameters from $\Theta$ resulting in a subnetwork parameterized by $\Theta'$, **4)** reset the remaining parameters in $\Theta'$ to their initial values in $\Theta_0$, which are referred to as the winning tickets $\hat{\Theta}$, **5)** fix the other parameters to zero and train $\hat{\Theta}$ only, we may attain a comparable level of accuracy with the trained-then-pruned model $\Theta'$. In that work, the third step is accomplished by simply preserving the parameters with the largest magnitude in $\Theta$. We found out that GSM can find a better set of winner tickets, as training the GSM-discovered tickets yields higher eventual accuracy than those found by magnitude (Table. 4). Concretely, we only replace step 3 by a pruning process via GSM on $\Theta$, and use the resulting non-zero parameters as $\Theta'$, and all the other experimental settings are kept the same for comparability. Interestingly, 100% parameters in the first fully-connected layer of LeNet-5 are pruned by 300× magnitude-pruning, such that the found winning tickets are not trainable at all. But GSM can still find reasonable winning tickets. More experimental details can be found in the codes.

Two possible explanations to the superiority of GSM are that **1)** GSM distinguishes the unimportant parameters by activation selection much earlier (at each iteration) than the magnitude-based criterion (after the completion of training), and **2)** GSM decides the final winning tickets in a way that is robust to mistakes (i.e., via activation re-selection). The intuition is that since we expect to find the parameters that have "won the initialization lottery", the timing when we make the decision should be closer to when the initialization takes place, and we wish to correct the mistakes immediately when we are aware of the wrong decisions. Frankle and Carbin also noted that it might bring benefits to prune as early as possible [15], which is precisely what GSM does, as GSM keeps pushing the unimportant parameters continuously to zero from the very beginning.

## 5 Conclusion

We proposed Global Sparse Momentum SGD (GSM) to directly alter the gradient flow for DNN pruning, which splits the ordinary momentum-SGD-based update into two parts: active update uses the gradients derived from the objective function to maintain the model's accuracy, and passive update only performs momentum-accelerated weight decay to push the redundant parameters infinitely close to zero. GSM is characterized by end-to-end training, easy implementation, lossless pruning, implicit connection rewiring, the ability to automatically discover the appropriate per-layer sparsity ratios in modern very deep neural networks and the capability to find powerful winning tickets.

## Acknowledgement

We sincerely thank all the reviewers for their comments. This work was supported by the National Key R&D Program of China (No. 2018YFC0807500), National Natural Science Foundation of China (No. 61571269, No. 61971260), National Postdoctoral Program for Innovative Talents (No. BX20180172), and the China Postdoctoral Science Foundation (No. 2018M640131). Corresponding author: Guiguang Ding, Jungong Han.

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
