[Reviews · NeurIPS 2019]

Reviewer 1



Update: Authors justified the choice of the competitor in empirical evaluation (thought it's better to add it to the body of the paper in camera ready if accepted). I find technique interesting, though i think results are exploratory and some-what preliminary, I think it's important for NeurIPS community to get familiar with these results. -------------- Authors suggest new gradient flow for prunning large DNN models. They identify and address major issues of current approaches, such as 1) prune then finetune for accuracy recover 2) prunning by custom learning (mostly custom regulizers). Authors introduce GSM - a new approach, that does not require finetuning afterwards and can be solved by means of vanilla SGD. GSM only updats the top Q values of the gradient based on the suggested metric (first order Taylor) --- |dL/dw * w|. This way weight decay gradually zero out all the redundunt parameters. Authors provide experimental study of their method. Strengths of the paper: - nice overview of the problem motivation (drawbacks of existent methods) - simple and straight-forward idea behind the algorithm Weaknesses of the paper: - no theoretical guarantees for convergence/pruning - though experiments on the small networks (LeNet300 and LeNet5) are very promising: similar to DNS [16] on LeNet300, significantly better than DNS [16] on LeNet5, the ultimate goal of pruning is to reduce the compute needed for large networks. - on the large models authors only compare GSM to L-OBS. No motivation given for the choice of the competing algorithm. Based on the smaller experiments it should be DNS [16], the closest competitor, rather than L-OBS, showed quite poor performance compared to others. - Authors state that GSM can be used for automated pruning sensitivity estimation. 1) While graphs (Fig 2) show that GSM indeed correlates with layer sensitivity, it was not shown how to actually predict sensitivity, i.e. no algorithm that inputs model, runs GSM, processes GSM result and output sensitivity for each layer. 2) Authors don't explain the detail on how the ground truth of sensitivity is achieved, lines 238-239 just say "we first estimate a layer's sensitivity by pruning ...", but no details on how actual pruning was done. comments: 1) Table 1, Table 2, Table 3 - "origin/remain params|compression ratio| non-zero ratio" --- all these columns duplicate the information, only one of the is enough. 2) Figure 1 - plot 3, 4 - two lines are indistinguishable (not even sure if there are two, just a guess), would be better to plot relative error of approximation, rather than actual values; why plot 3, 4 are only for one value of beta while plot 1 and 2 are for three values? 3) All figures - unreadable in black and white 4) Pruning majorly works with large networks, which are usually trained in distributed settings, authors do not mention anything about potential necessity to find global top Q values of the metric over the average of gradients. This will potentially break big portion of acceleration techniques, such as quantization and sparsification.

Reviewer 2



Originality: This paper has two major drawbacks in its originality segment: 1) the field of NN-pruning is quite busy with many related papers populating the field and 2) it does not compare against the following very similar paper: Faster gaze prediction with dense networks and Fisher pruning by Theis et al 2018. This paper uses the fisher information to prune features during gradient descent subject to user-preset-computational constraints. Quality: The paper is technically interesting, but makes one leap which is unclear: the authors claim to be model agnostic and instead to be putting all their assumptions into the SGD method. However, the curvature calculations (via Taylor approximations) are model-dependent and actually exploit model structure to determine if a weight should be pruned. It would be great to relate this to the Hessian and the Fisher Information (see: Fisher pruning) to clarify the relationship to the model. Apart from that, another drawback of the paper is the need to express the compression ratio, which is quite an unnatural quantity to have to hand-specify and is not really what a user wants to control. Constraints typically exist in speed or memory space, not in compression ratios. The experiments are pretty well executed, I particularly enjoyed the study of the feature-re-activation, which studies a specific property of this model. Clarity: The paper is well written and concise. Significance: This paper manages to not need complex criteria or multi-stage models to achieve its goal of sparsifying. In the long term, this can be an important property to make pruning a pragmatic modeling tool enshrined in software.

Reviewer 3



(1) This paper is well written. (2) To my knowledge, most of the preceding methods only prune relatively shallow models like Alexnet and Vgg, where it is possible to manually set the layer-wise pruning rates based on trial-and-error. But the proposed method requires no pre-defined layer-wise pruning rates, which is especially good on very deep models. (3) The proposed method (GSM) achieves lossless pruning. Compared to the classic L1/L2-based pruning method [Han et al. Learning both ...], which use L1/L2 regularization to reduce the magnitude of parameters (at the cost of compromised accuracy) and then prune the parameters (with accuracy reduction again), the model encounters no accuracy drop when pruned after GSM training. (4) The proposed method is intuitive and easy to understand. The method utilizes momentum in a natural and creative way: to accelerate the process of a parameter moving towards a constant direction. (5) The main reasons for me to vote for accepting the paper are the novelty and potential insights. The idea of directly modifying the gradients to accomplish a certain task is intriguing. Actually, we always customize the loss function to indirectly modify the gradients which control the direction of training, but rarely directly transform the gradients.

[Author Response · NeurIPS 2019]

**To Reviewer #1. L-OBS as competitor.** Though the very deep models are widely utilized in real-world applications, they were rarely chosen by the previous pruning methods for the effectiveness validation, as it is difficult to manually tune the layer-wise pruning ratios to achieve a satisfactory compression ratio. We chose L-OBS as the competitor on ResNet-50 simply because 1) it is the only previous method which reported results on ResNet-50, to the best of our knowledge, and 2) as DNS requires some hyper-parameters, it is unpractical for us to tune the method carefully on ResNet-50 and reproduce a reasonable performance. **Pruning sensitivity.** The sensitivity is *not predicted by GSM*, but we noticed that if directly pruning a layer of the original model by a certain ratio (with all the other layers kept the same) results in a significant accuracy reduction, GSM chooses to prune it less, and vice versa. This is actually a strength of GSM and a quantitative validation is provided in Section 4.2. The sensitivity of a layer is an underlying property which can be defined via a natural proxy: the accuracy reduction caused by pruning a certain ratio of parameters. In our paper, we first evaluate such sensitivity via single-layer pruning attempts with different pruning ratios (Figure 2). E.g., for the curve labeled as "prune 90%" of LeNet-5, we first experiment on the first layer by setting 90% of the parameters with smaller magnitude to zero, and testing the model to obtain the accuracy. Then we restore the first layer, prune the second layer and test. The same procedure is applied to the third and fourth layers. After that, we vary the pruning ratio and obtain three curves in the same way. From such experiments we learn that the third layer is the least sensitive. Then we show the layer-wise sparse ratios of the GSM-pruned model. Naturally, for a sensitive layer, GSM is expected to prune it less. I.e., when using GSM, the sensitivity of a layer can be measured using another proxy: the resulting sparse ratio of the layer. As shown in Figure 2, the sensitivities measured in the two proxies are closely related, thus we conclude that GSM automatically knows which layers to prune harder. We will provide more details in the final version. **About Figures.** We use 3 values of $\beta$ to show a larger $\beta$ can accelerate the reduction of parameter value, and use $\beta = 0.98$ as the representative to show that the approximation is accurate enough. We will plot the relative approximation error of the 3 values in the final version, and use markers on curves to make figures readable in black and white.

**To Reviewer #2. About compression ratio.** DNN *connection pruning* (e.g., our method) targets at significantly reducing the number of non-zero parameters, resulting in a sparse model, which can be stored using much less space, but cannot effectively reduce the computational burdens on off-the-shelf hardware and software platforms. On the other hand, *channel pruning* (e.g., Fisher Pruning) cannot achieve a high compression ratio of the model size, but can convert a wide CNN into a narrower (but still dense) one to reduce the runtime memory space and accelerate the computation. Connection pruning and channel pruning are complementary and often used together. For connection pruning, the core trade-off is the model size v.s. accuracy. E.g., the storage space of a mobile APP is usually constrained. Therefore, with a high compression ratio, we can use a big and sparse model to achieve better accuracy than a small and dense one. We will mention this in the final version. **Fisher Pruning.** Our method differs from Fisher Pruning in two aspects. 1) Fisher Pruning falls into the category of channel pruning whereas ours is a typical connection pruning method. 2) The metrics used to evaluate a parameter are different. Fisher Pruning measures the importance of a single parameter $\theta$ using a derivation of [Molchanov et al. Pruning convolutional neural networks for resource efficient inference]: $T'(\theta) = \frac{1}{2N}\theta^2 \sum_{n=1}^{N} g_n^2$, where $N$ is the number of data points for measurement, and $g_n$ is the gradient of $\theta$ on the $n$-th data point. Fisher Pruning uses this to *greedily* remove parameters *one-by-one*. On the other hand, we select the important parameters by $T(\theta) = |g\theta|$ for each batch of data. **Model agnostic.** GSM is model agnostic because it makes no assumptions on the model structure or the form of loss function. All the information GSM needs is the value of gradient and parameter. The information of model structure is actually encoded in the gradients via back propagation. I.e., the calculation of gradients is model-related, of course, but it is model-agnostic to use them for GSM pruning. **Lottery ticket.** Thanks for your suggestion, we found out that our method can be used as a more powerful method to find the winning tickets! The major contribution of the lottery paper is an observation (rather than a new method) that the important parameters which are trained to become important (winning tickets) are actually important at the very beginning (after random initialization but before training). That paper discovered that if we 1) randomly initialize a network parameterized by $W_0$, 2) train and obtain $W$, 3) prune some parameters based on the properties of $W$ resulting in a subnetwork parameterized by $W' \subset W$, which is referred to as the winning tickets, 4) find the parameters $\hat{W} \subset W_0$ in the initialized model corresponding to $W'$, 5) train $\hat{W}$ only, and remove the other parameters, we may obtain a comparable level of accuracy with the trained-then-pruned model $W'$. In that paper, the third step is accomplished by simply preserving the parameters with the largest magnitude. In our experiments, we found out that GSM can find a better set of winner tickets than the original simple magnitude-based method. Concretely, we only replace the third step by a pruning process via GSM, and use the resulting non-zero parameters as the winning tickets, and all the other training settings are kept the same as introduced in our paper. On LeNet-300-100 with a compression ratio of $60\times$, finding winning tickets by the original magnitude criterion and GSM delivers a final accuracy of 96.85% and 97.36%. On LeNet-5 with a compression ratio of $300\times$, the accuracies of the two methods are 97.94% and 99.04%.

**To Reviewer #3.** GSM can be viewed as sampling and training a subnetwork of the model. As many classic works have shown, such a subnetwork can satisfactorily approximate the original model. Though GSM focuses on global pruning, we still performed layer-wise pruning experiments, and the results are better than the pruning-then-finetuning methods, which are not demonstrated due to page space limitation.

[Meta-Review · NeurIPS 2019]

The paper proposes a method for pruning deep networks based on the largest values of the gradient vector. The idea is new compared to previous attempts; although it is somewhat related to Fisher pruning, that is also based on magnitudes of gradients, the method here is more of an SGD variant rather than a post-training evaluation method. The techniques do not come with rigorous guarantees, but the reviewers agree that the experiments and surrounding studies are interesting enough to incite future research around this method. Based on this, I believe the paper would be a appreciated by the attendees of NeurIPS The idea is to update only the values having the maximum gradient magnitude Though the results are somewhat exploratory and preliminary, their potential is great and their novelty is significant enough to be presented in NeurIPS